# Mapping the Mangrove Forest Canopy Using Spectral Unmixing of Very High Spatial Resolution Satellite Images

**Florent Taureau** [1,*] **, Marc Robin** [1]**, Christophe Proisy** [2,3]**, François Fromard** [4]**, Daniel Imbert** [5] **and Françoise Debaine** [1]

[1] Université de Nantes, UMR CNRS 6554 Littoral Environnement Télédétection Géomatique, Campus Tertre, 44312 Nantes, France; marc.robin@univ-nantes.fr (M.R.); francoise.debaine@univ-nantes.fr (F.D.)

[2] AMAP, IRD, CNRS, CIRAD, INRA, Université de Montpellier, 34000 Montpellier, France; christophe.proisy@ird.fr

[3] French Institute of Pondicherry, Pondicherry 605001, India

[4] ECOLAB, Université de Toulouse, CNRS, INPT, UPS, 118 route de Narbonne, 31500 Toulouse, France; francois.fromard@univ-tlse3.fr

[5] UMR Ecologie des Forêts de Guyane (EcoFoG), INRA, CNRS, Cirad, AgroParisTech, Université des Antilles, Université de Guyane, 97310 Kourou, French Guiana, France; daniel.imbert@univ-ag.fr

* Correspondence: florent.taureau@univ-nantes.fr; Tel.: +33-253-487-652

**Abstract:** Despite the low tree diversity and scarcity of the understory vegetation, the high morphological plasticity of mangrove trees induces, at the stand level, a very large variability of forest structures that need to be mapped for assessing the functioning of such complex ecosystems. Fully constrained linear spectral unmixing (FCLSU) of very high spatial resolution (VHSR) multispectral images was tested to fine-scale map mangrove zonations in terms of horizontal variation of forest structure. The study was carried out on three Pleiades-1A satellite images covering French island territories located in the Atlantic, Indian, and Pacific Oceans, namely Guadeloupe, Mayotte, and New Caledonia archipelagos. In each image, FCLSU was trained from the delineation of areas exclusively related to four components including either pure vegetation, soil (ferns included), water, or shadows. It was then applied to the whole mangrove cover imaged for each island and yielded the respective contributions of those four components for each image pixel. On the forest stand scale, the results interestingly indicated a close correlation between FCLSU-derived vegetation fractions and canopy closure estimated from hemispherical photographs ($R^2 = 0.95$) and a weak relation with the Normalized Difference Vegetation Index ($R^2 = 0.29$). Classification of these fractions also offered the opportunity to detect and map horizontal patterns of mangrove structure in a given site. K-means classifications of fraction indeed showed a global view of mangrove structure organization in the three sites, complementary to the outputs obtained from spectral data analysis. Our findings suggest that the pixel intensity decomposition applied to VHSR multispectral satellite images can be a simple but valuable approach for (i) mangrove canopy monitoring and (ii) mangrove forest structure analysis in the perspective of assessing mangrove dynamics and productivity. As with Lidar-based surveys, these potential new mapping capabilities deserve further physically based interpretation of sunlight scattering mechanisms within forest canopy.

**Keywords:** mangrove; Remote Sensing; forest structure; hemispherical photographs; Guadeloupe; Mayotte; New Caledonia

## 1. Introduction

Mangroves are tropical forest ecosystems characterized by trees and shrubs adapted to a high level of salinity and growing in the tidal zone [1]. The structure of mangrove vegetation may vary a lot from place to place ranging from very dense forest stands to sparse shrubs according to regional geomorphic settings, local hydrology and topography, and site history [2,3]. In most mangrove areas, seaward/landward environmental gradients lead to an organization of the vegetation more or less parallel to the seashore or riversides, the so-called "zonation" pattern [4].

Despite increasing awareness of the myriad services provided by mangroves [5], mangroves are threatened on the global scale [6–11]. Their global extent decreased by about 25% from 1980 to 2005 [12], and their rate of extinction is estimated to be higher than tropical rainforests [12,13]. Mangroves are also affected by climate change [14–17] and human activities [18].

Fine-scale monitoring of mangrove ecosystems over large areas are crucial for national or international observatories. Those observatories need robust mangrove maps based on vegetation typologies which are built on both species composition and structural properties of stands. Characterizing the horizontal structure (zonation) of mangrove vegetation [19] proved essential for the detection of changes in mangroves submitted to natural or anthropogenic disturbances [20,21]. Such zonation can be revealed by canopy organization. Forest canopy is the highest vegetation surface that "follows the irregular contour of the upper tree crowns" [20]. It is composed of tree leaves, branches, fruits and flowers as well as the gaps between and within the tree crowns [22]. The structure of the mangrove canopy evolves in time and space as a resulting component of multiple factors like species-specific crown morphology, total irradiance, soil conditions, site history, and forest dynamics [23].

Satellite or aerial photographs deliver two-dimensional images of forest canopy in which both spectral and textural signatures are influenced not only by the optical properties but also by the distributional patterns of leaves, branches and trunks [24]. Assessing mangrove canopy properties from multispectral (MS) satellites images relies on various methods such as vegetation indices, particularly the Normalized Difference Vegetation Index [25–27]. This index is often transformed in several classes of canopy structures (e.g., "dense vegetation" for NDVI > 0.7, "sparse vegetation" for NDVI < 0.4). However, many studies have shown that NDVI is limited by the reflectance of background surfaces such as soil and water [28]. Complementary to the use of NDVI is spectral mixture analysis [29–32]. Studies have used Linear Spectral Unmixing (LSU) algorithm to assess vegetation organization from moderate to high resolution datasets such as those provided by Landsat, Aster, SPOT, and, more recently, Sentinel sensors [33–35]. The unmixing process attempts to separate ground contributions of water, soil and shadow reflectance from pure vegetation reflectance [36] in order to (i) quantify modelled mangrove canopy closure and (ii) classify mangrove types according to the four unmixed components. This decomposition is potentially useful for the analysis of canopy structural properties [37] and particularly advocates for the use of very high spatial resolution (VHSR) satellite images. Even in small pixels with a size of about one square meter, intensity is often "mixed" i.e., dependent on the proportion and arrangement of soil and vegetation components illuminated by sunlight [38]. This also affects any vegetation indices but, on the other hand, may provide valuable information on spatial patterns of forest zonations characterized by distinct flooding levels, forest height, leaf area index, or gaps within the tree crowns.

Even if increasing availability of repeatable VHSR multispectral (MS) satellite images at affordable cost allows the development of new and promising methods based on spectral and textural analysis [39–42]. Surprisingly, only a few studies have applied LSU on VHSR satellite images even though they provide promise for assessing canopy properties [42]. For example, flooded mangroves may have a dominant water fraction, while soil fraction on saltpan areas will be dominant. Moreover, the shadow fraction may be of importance in tall forest depending on the sun elevation and viewing angles.

This paper aims at exploring the potential of FCLSU to (i) characterize mangrove canopy density derived from vegetation fraction and (ii) to therefore map spatial patterns of mangrove zonations in

three different sites using the same approach and the same forest structure typology. It is worth noting that vegetation fraction was compared to canopy closure estimates from hemispherical photographs analysis. Future research perspectives in both remote sensing and mangrove forest ecology are then discussed.

## 2. Materials

### 2.1. Study Sites

Three sites in the Indian, Atlantic, and Pacific oceans were chosen to test the spatial reproducibility of the FCLSU approach over a broad range of forest structures.

The Guadeloupe archipelago includes about 3200 hectares of mangroves mainly located in the region of Grand-Cul-de-Sac Marin Bay (16.312934°N; 61.57608°W). The bay is sheltered from ocean currents and waves by a 30-km-long coral reef but can be severely impacted by strong winds generated by climatic events such as tropical storms and hurricanes [43]. Because of increasing land use activities such as urbanization and agriculture, the regulations issued by the National Park of Guadeloupe and the French Coastal Conservancy require baseline data and monitoring methods to assess or even prevent mangrove disturbances. *Avicennia germinans*, *Rhizophora mangle*, and *Laguncularia racemosa* are the dominant species, while *Conocarpus erectus* and *Avicennia schaueriana* can also be found.

Mayotte is an insular French department (374 km$^2$) located in the Northern Mozambique Channel in the Indian Ocean. This volcanic island has 650 hectares of mangroves, mainly located at the bottom of bays and protected by a 1500 km$^2$ continuous coral reef [44]. In the south of the island (12.923852°S; 45.150654°E), the extent of the Bouéni Bay mangroves reached about 182 ha in 2009, as indicated in [45]. The mangrove surface area is currently decreasing (−4.54 % from 2003 to 2009) because of the combined effects of coastal erosion [46] and increased anthropogenic pressures directly linked to high population growth and very rapid economic development [44,45]. The Bouéni Bay mangroves have been under the protection of both the French Coastal Conservancy since 2007 and the regulations issued by the Marine National Park of Mayotte created in 2010. About ten mangrove species can be found. Among the dominant ones are *Avicennia marina*, *Lumnitzera racemosa*, *Bruguiera gymnorhiza*, *Ceriops tagal*, *Rhizophora mucronata*, and *Sonneratia alba*, while *Pemphis acidula*, *Xylocarpus granatum*, *Xylocarpus moluccensis*, and *Heritiera littoralis* are less common.

New Caledonia is a southern island of the Pacific Ocean (Melanesia), located about 1500 km from the east coast of Australia. The study area in Chasseloup Bay is located on the northwest coast (20.934534°S; 164.655199°E) in the Northern Province. In this region, about 1367 ha of mangroves develop along meanders fed by fresh waters from the Temala River, Wepannook River and Cemetery Creek. The delta is protected from offshore waves by large coral reefs but inland it is worth noting that a shrimp farm operates less than two kilometers upstream and near a landfill. Eight mangrove species can be found [47]: *Acanthus ilicifolius*, *Avicennia marina*, *Bruguiera gymnorhiza*, *Excoecaria agallocha*, *Lumnitzera racemosa*, *Rhizophora samoensis*, *Rhizophora X selala*, and *Rhizophora stylosa*, with *Rhizophora* sp. and *Avicennia marina* representing the most abundant species.

### 2.2. Forest Data

Field experiments were carried out in February, March and May 2014 in Guadeloupe, Mayotte and New Caledonia, respectively. Forest areas for field sampling were selected from the visual analysis of satellite images presented in Figure 1 and delineated in situ with the objective of capturing a gradient of canopy closure within a given study area. The geographic position was recorded using standard GPS surveys. The resulting sampling areas varied from 25 m$^2$ to 625 m$^2$ relative to the forest stand structure (i.e., the number and size of trees). The final data set included 10 plots in Guadeloupe, 13 in Mayotte, and 12 in New Caledonia (Table 1). After species identification of each sampled tree, the diameter at breast height (DBH) was measured, the number of stems per area unit was counted,

plot basal area values were computed, and the mean canopy height formed by dominating species was evaluated with a hand-held TruePulse® rangefinder.

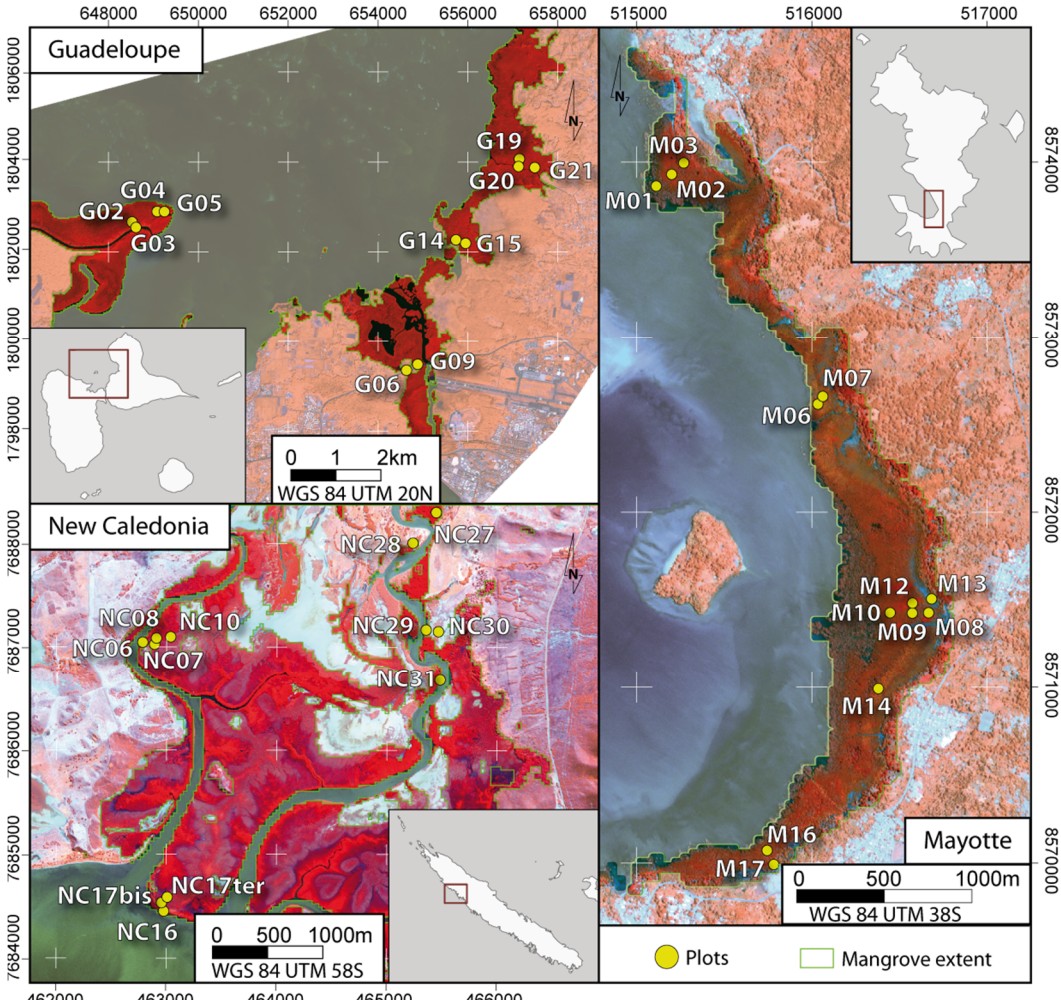

**Figure 1.** Study sites and location of plots in the three islands (French Overseas Territories) based on Pleiades 1B images.

### 2.3. Hemispherical Photographs

Three series of digital hemispherical photographs (HP) were taken using a GoPro Hero-3 and sensor sensitivity ranged between ISO 100 and 400. Image size was 4000 × 3000 pixels. Hemispherical photographs were obtained with a focal length of 14 mm opening at an angle of view of 149.2°. The shutter speed was adjusted depending on light conditions (between 1/100 and 1/500s). All photographs were taken from the forest plot center. Approximatively 15 to 20 photographs were taken for each plot with a one second delay between each photo in order to have same light exposition condition for each photo. Then the best photograph i.e., the photograph which is not affected by sunlight saturation or branches movement caused by wind has been kept and used for analysis.

The HP analysis consisted of evaluating the ratio between the vegetal and the free sky image pixels [48] using binarization. The Object Based Image Analysis (OBIA) classification process implemented in the eCognition© software was used [49–51]. Each photograph was divided into several objects. Each object was then classified as sky or vegetation. Objects were determined using the "Split Contrast" segmentation algorithm, which executes several chessboard segmentations at different scales before separating the objects into two categories—light and dark—from a threshold value set after several iterations that maximizes their contrast [52]. The ratio between the number of

pixels classified as "vegetation" and the total number of pixels of the image gave an estimate of the canopy closure of the forest plot considered.

**Table 1.** Ground data with the main forest parameters and canopy closure estimated from hemispherical photographs (HP) for all plots in the three sites (G: Guadeloupe; M: Mayotte; NC: New Caledonia). The DBH and basal areas represented by "-" are for multi-stem trees. Cluster ID column must be interpreted after the reading of the method and result sections. R = *Rhizophora* sp.; A = *Avicennia* sp.; Lr = *Laguncularia racemosa*; B = *Bruguiera* sp.; E = *Excoecaria agallocha*; Lu = *Lumnitzera racemosa*; Sa = *Sonneratia alba*; C = *Ceriops* sp.

| Plot ID. | DBH (cm) | Stem Density (Trees.ha$^{-1}$) | Basal Area (m$^2$.ha$^{-1}$) | Mean Height (m) | Dominating Species | Canopy Closure (from HP) | Vegetation Fraction | Water Fraction | Soil Fraction | Shadow Fraction | Cluster ID |
|---|---|---|---|---|---|---|---|---|---|---|---|
| G02 | 13 | 1825 | 31 | 4 | A+R | 0.82 | 0.68 | 0.21 | 0.00 | 0.09 | 4 |
| G04 | 7 | 1500 | 9 | 3 | A+R | 0.61 | 0.64 | 0.19 | 0.00 | 0.14 | 2 |
| G05 | 10 | 3900 | 34 | 6 | R | 0.82 | 0.78 | 0.18 | 0.00 | 0.03 | 2 |
| G06 | 7 | 3700 | 19 | 4 | Lr | 0.54 | 0.51 | 0.23 | 0.02 | 0.21 | 4 |
| G09 | 9 | 2100 | 28 | 4 | R | 0.71 | 0.64 | 0.26 | 0.00 | 0.08 | 2 |
| G14 | 5 | 3200 | 13 | 2 | R | 0.8 | 0.78 | 0.12 | 0.09 | 0.00 | 5 |
| G15 | 9 | 3600 | 25 | 7 | R+A | 0.77 | 0.80 | 0.10 | 0.04 | 0.04 | 5 |
| G19 | 6 | 1475 | 7 | 4 | Lr | 0.67 | 0.65 | 0.21 | 0.00 | 0.13 | 2 |
| G20 | 6 | 4125 | 19 | 4 | A+Lr | 0.76 | 0.77 | 0.08 | 0.03 | 0.11 | 1 |
| G21 | 12 | 2400 | 34 | 9 | A+R | 0.7 | 0.60 | 0.16 | 0.00 | 0.22 | 4 |
| M01 | - | 300 | - | 5 | Sa | 0.26 | 0.34 | 0.08 | 0.23 | 0.33 | 5 |
| M02 | 23 | 1400 | 62 | 7 | R | 0.74 | 0.74 | 0.00 | 0.01 | 0.23 | 1 |
| M03 | 21 | 1500 | 90 | 8 | A+R | 0.74 | 0.73 | 0.00 | 0.01 | 0.24 | 1 |
| M06 | 9 | 4300 | 51 | 5 | C+B+R | 0.72 | 0.70 | 0.03 | 0.12 | 0.13 | 3 |
| M07 | 4 | 31,746 | 45 | 3 | C+B+R | 0.77 | 0.70 | 0.02 | 0.06 | 0.19 | 3 |
| M08 | 4 | 46,031 | 57 | 2 | C | 0.6 | 0.48 | 0.05 | 0.18 | 0.27 | 5 |
| M09 | 11 | 1244 | 19 | 4 | R+C | 0.66 | 0.57 | 0.07 | 0.08 | 0.26 | 2 |
| M10 | 11 | 888 | 11 | 7 | R+B+C | 0.58 | 0.62 | 0.0 | 0.03 | 0.29 | 2 |
| M12 | 3 | 3100 | 49 | 5 | R+C+B | 0.92 | 0.93 | 0.00 | 0.00 | 0.06 | 1 |
| M13 | - | 225 | - | 4 | A | 0.22 | 0.24 | 0.06 | 0.50 | 0.18 | 4 |
| M14 | 10 | 2755 | 36 | 4 | C+B+R | 0.82 | 0.79 | 0.02 | 0.06 | 0.12 | 3 |
| M16 | 26 | 1200 | 64 | 10 | R | 0.77 | 0.71 | 0.05 | 0.02 | 0.19 | 1 |
| M17 | - | 250 | - | 4 | A | 0.31 | 0.39 | 0.05 | 0.06 | 0.48 | 5 |
| NC06 | 20 | 1000 | 33 | 5 | R | 0.79 | 0.61 | 0.00 | 0.00 | 0.36 | 1 |
| NC07 | 1 | 18,400 | 2 | 1 | A | 0.27 | 0.22 | 0.01 | 0.18 | 0.57 | 6 |
| NC08 | 2 | 4900 | 2 | 1 | R | 0.38 | 0.29 | 0.04 | 0.14 | 0.51 | 3 |
| NC10 | 4 | 825 | 2 | 2 | R | 0.2 | 0.25 | 0.01 | 0.06 | 0.65 | 4 |
| NC16 | 10 | 2600 | 22 | 3 | R+A | 0.43 | 0.39 | 0.07 | 0.03 | 0.49 | 3 |
| NC17bis | 2 | 24,000 | 5 | 1 | A | 0.27 | 0.22 | 0.05 | 0.18 | 0.53 | 6 |
| NC17ter | 6 | 1422 | 4 | 2 | R | 0.48 | 0.35 | 0.00 | 0.04 | 0.59 | 4 |
| NC27 | 18 | 800 | 23 | 6 | B + R | 0.73 | 0.70 | 0.00 | 0.00 | 0.29 | 2 |
| NC28 | 13 | 500 | 7 | 6 | E | 0.68 | 0.62 | 0.03 | 0.01 | 0.32 | 1 |
| NC29 | 7 | 1350 | 16 | 3 | B | 0.56 | 0.40 | 0.02 | 0.10 | 0.46 | 4 |
| NC30 | 6 | 500 | 1 | 3 | Lu | 0.4 | 0.35 | 0.13 | 0.22 | 0.29 | 3 |
| NC31 | 12 | 1400 | 18 | 8 | R | 0.81 | 0.81 | 0.12 | 0.03 | 0.02 | 2 |

*2.4. Satellite Image Acquisition and Preprocessing*

Pleiades-1B satellite images were acquired in each study area in a Geotiff format (Table 2) at level 1C. All images were acquired near to high tide time. However, it was difficult to find cloud-free images with similar viewing and sun angles. We acknowledge that the retrieval of canopy properties can depend on viewing geometry configurations. For example, acquisition angles can marginally affect the estimation of canopy closure, which can be overestimated with a high view angle [53], while a frontward sun-viewing configuration can affect the spectral surface reflectance [54]. To avoid this problem as much as possible, the images with the smallest angle were favoured, which was not always possible given the cloud cover and the desired dates of acquisition.

The intensity of the satellite image pixels was then transformed into radiance values $L_\lambda$ (mW.cm$^2$.sr$^1$.µm$^{-1}$) using the sensor specific calibration values provided by Airbus Defense and Space. Then the radiance of each pixel was converted into the top-of-atmosphere (TOA) reflectance $\rho$ using the following equation (Equation (1)):

$$\rho_\lambda = \frac{\pi.L_\lambda.D^2}{E_{sun}.\cos\theta_s} \qquad (1)$$

where $D$ is the sun-earth distance (expressed in astronomical units), $E_{sun}$ is the corresponding mean solar exoatmospheric spectral irradiance, and $\theta_s$ is the solar zenithal angle.

**Table 2.** Main acquisition parameters of multispectral (MS) Pleiades satellite images. $\theta_s$ and $\theta_v$ are the zenithal sun and viewing angles, respectively, while $\phi_{s-v}$ is the sun-sensor absolute difference azimuthal angle.

| Site | Sensor | Bands Used | Pixel Size | Acquisition Date | High Tide Time | $\theta_s$ (°) | $\theta_v$ (°) | $\phi_{s-v}$ (°) |
|------|--------|------------|------------|------------------|----------------|----------------|----------------|------------------|
| Guadeloupe | 1B | MS | 2 m | 14 December 2013 | 2 h before | 43 | 9 | 25 |
| Mayotte | 1B | MS | 2 m | 18 April 2013 | 2 h after | 33 | 20 | 135 |
| New Caledonia | 1B | MS | 2 m | 27 June 2013 | 1 h before | 51 | 4 | 149 |

Atmospheric corrections were achieved using the FLAASH module process implemented in the ENVI 5.1®software, this module being based on the MODTRAN radiative transfer model [55].

## 3. Methods

As shown in Figure 2, the method consisted of three steps. First, satellite images were analyzed according to the FCLSU method. Then, vegetation fractions were compared to canopy closure estimations from analyses of HP through linear regressions. Lastly, the vegetation structure was mapped across a k-mean classification of all fractions resulting from the FCLSU process.

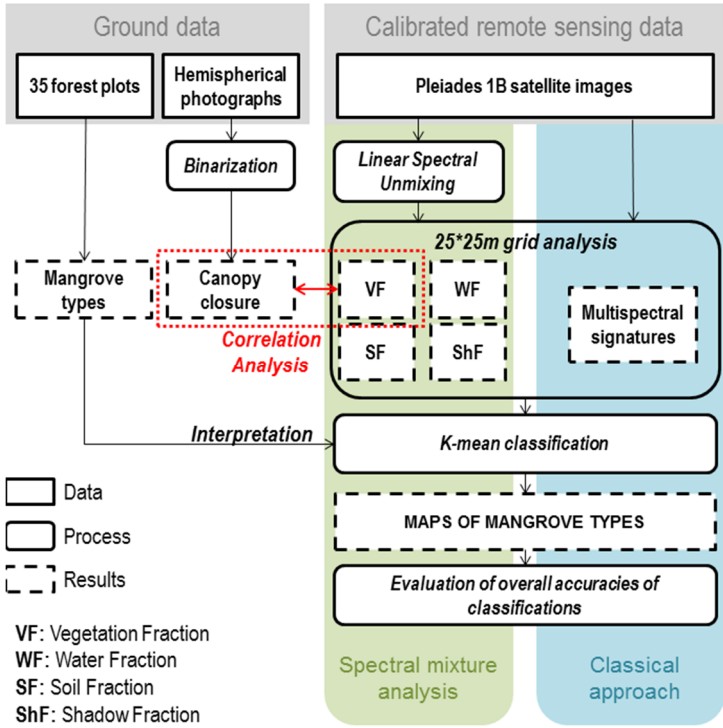

**Figure 2.** Flowchart of data processing in mangrove mapping.

### 3.1. Fully Constrained Linear Spectral Unmixing (FCLSU)

The algorithm progresses in two steps: first, the endmember selection, which consists of selecting different pure spectra representative of the different surfaces composing the image, and then the decomposition of pixel reflectance in order to quantify the fraction of endmembers within the pixel.

### 3.1.1. Endmember Selection

Spectral mixture analysis assumes that pixels can be modeled as a linear combination of the spectral contribution of spectrally pure land cover components called endmembers [33]. To study the mangrove canopy, the pixel intensity was decomposed into four endmembers: vegetation, water, soil, and shadow. The spectral satellite multispectral signature of each endmember was derived from the average reflectance over more than 10 pixels on MS bands at native spatial resolution (2 m). The vegetation endmember was obtained over dense mangrove areas showing the highest degrees of canopy closure, the water spectra were taken from non-turbid water in order to avoid the mixing with soil spectral properties that occurs with turbid water, and the soil spectra were selected on dry bare soil in order to avoid the mixing with water that occurs with wet soils. Shadow spectra were taken from soil surfaces shaded by high mangrove trees and expanded over more than nine pixels [56]. These spectra composed the spectral library used to set the FCLSU algorithm.

### 3.1.2. Decomposition of Pixel Spectra according to Endmember Spectral Contributions

The decomposition of pixels consisted of evaluating the fractional cover of endmembers of each pixel according to the following equation (Equation (2)):

$$R_k = \sum_{i}^{n} a_i E_{i,k} + \xi_k \tag{2}$$

with $R_k$ = pixel reflectance value at $k$ wavelength, $a_i$ = abundance of endmember $i$, $E_{i,k}$ = reflectance of endmember $i$ at $k$ wavelength, $\xi_k$ = error at $k$ wavelength, and $n$ = number of endmembers.

The results of FCLSU, called "fractions," are proportional to the contribution of each "endmember" in the global spectral signature of the pixel. FCLSU takes into account two additional major constraints (Equation (3)): the sum of the endmember fractions within one pixel must be equal to 1 (sum constraint) and these fractions must be positive (non-negativity constraint). These constraints enable interpretable fraction values to be obtained [57].

$$\sum_{j=1}^{n} \alpha_i = 1 \\ \text{with } 0 \le \alpha_i \le 1, \ for \ 1 \le n \le k \tag{3}$$

and with $n$ = number of endmembers, $\alpha$ = fraction of endmembers $i$, and $k$ = number of spectral bands.

Applying this algorithm to satellite images with the selected endmembers provided four images representing vegetation (VF), water (WF), soil (SF), and shadow (ShF) fractions, respectively. The Sentinel Application Platform software (SNAP v.5.0.0) was used to perform the FCLSU.

### 3.2. Validation of the FCLSU

A grid composed of 25 m × 25 m polygons was built on unmixed images to fit the swath captured by hemispherical photographs. Mean values of all fractions were calculated for all polygons of the grid.

In order to validate the model, polygons were selected corresponding to the HP location. Values of mean VF were then compared with canopy closure estimated from HP. NDVI mean values of polygons were also compared with canopy closure from HP.

### 3.3. Vegetation Structure Mapping

The potential of unmixing four components from satellite images of mangroves to characterize vegetation structure was then analyzed. To do so, a classification applied on the mean fraction values (fraction classification) was compared with a classification applied on the mean spectral values (spectral classification) calculated inside all the polygons of the 25 × 25 m grid.

Both classifications were performed with a 6-cluster K-mean classification. The number of clusters was chosen according to two indices: explained inertia and Calinski-Harabzs index [58] presented in Figure 3.

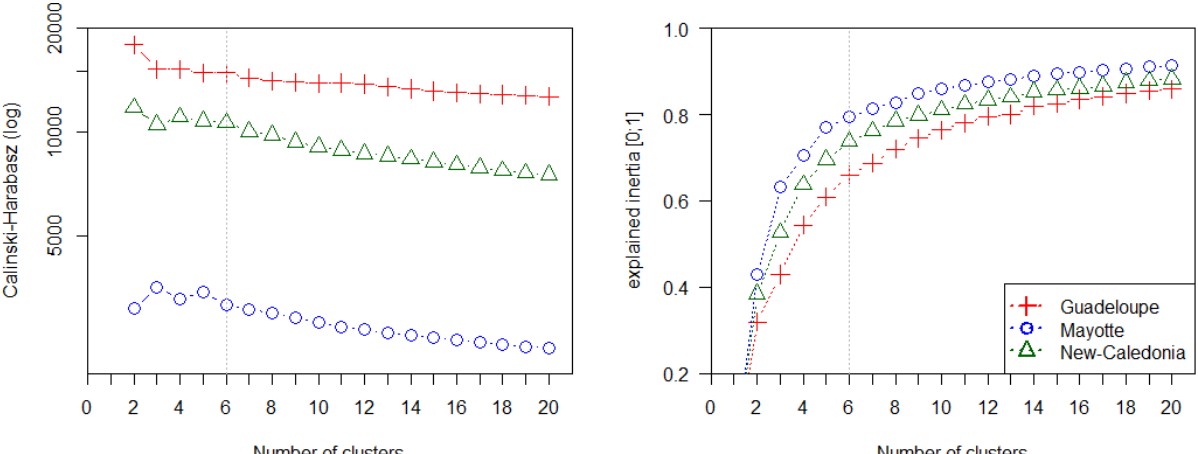

**Figure 3.** Calinski-Harabasz index and explained inertia values according to different number of cluster.

Classifications were then evaluated with confusion matrices built with six photo-interpreted samples composed of one hundred polygons per class. The results are presented in the following section.

## 4. Results

### 4.1. Performance of the FCLSU Approach for Characterizing Mangrove Forest Structures

Average values of the four FCLSU endmembers were compared to canopy closure estimates from hemispherical photographs and the Normalized Difference Vegetation Index over the 35 forest plots (Table 1; Figure 4). The FCLSU-derived vegetation fraction was first tested before the potential of the full decomposition was examined for mangrove forest mapping.

Canopy closure estimates and vegetation fraction values were strongly and linearly correlated (Figure 4, top left) with $R^2 = 0.91$ (*p*-value = $2.2 \times 10^{-16}$) over the image dataset and forest plots. In addition, a Shapiro-Wilk test of normality on residues between the previous relationship and the extracted values of VF (W = 0.982, *p*-value = 0.815) showed a normal distribution suggesting that extreme values did not influence the relationship too significantly. NDVI was poorly correlated with the vegetation fraction (bottom left) with $R^2 = 0.34$ (Figure 4) and also with the canopy closure ($R^2 = 0.29$).

Graphs of the regressions showed that the relationship between CC and VF was linear and near the 1:1 line, regardless of the canopy structure. The relationship between CC and NDVI was near the 1:1 line when the canopy closure reached high values (more than 0.5). The same observation was made for the VF and NDVI regression line. For open canopies with a low value of CC, the relationship between CC and NDVI clearly became non-linear and deviated from the 1:1 line. This tends to demonstrate that the NDVI is dependent on the canopy structure; more precisely, with an open canopy the NDVI is very inconstant.

The variability of FCLSU vegetation fractions, canopy closure, and NDVI values was analyzed between and across regional sites (Figure 5). Overall, FCLSU vegetation fractions and canopy closure exhibited similar variability patterns, distinct from those of the NDVI signatures. Estimates of canopy closure and vegetation fraction for all plots varied from 0.20 to 0.92 and from 0.24 to 0.93, respectively. The NDVI responses of mangrove forest plots remained between 0.59 and 0.97. This result suggests a lack of sensitivity of NDVI but a close match between VF and canopy closure estimates.

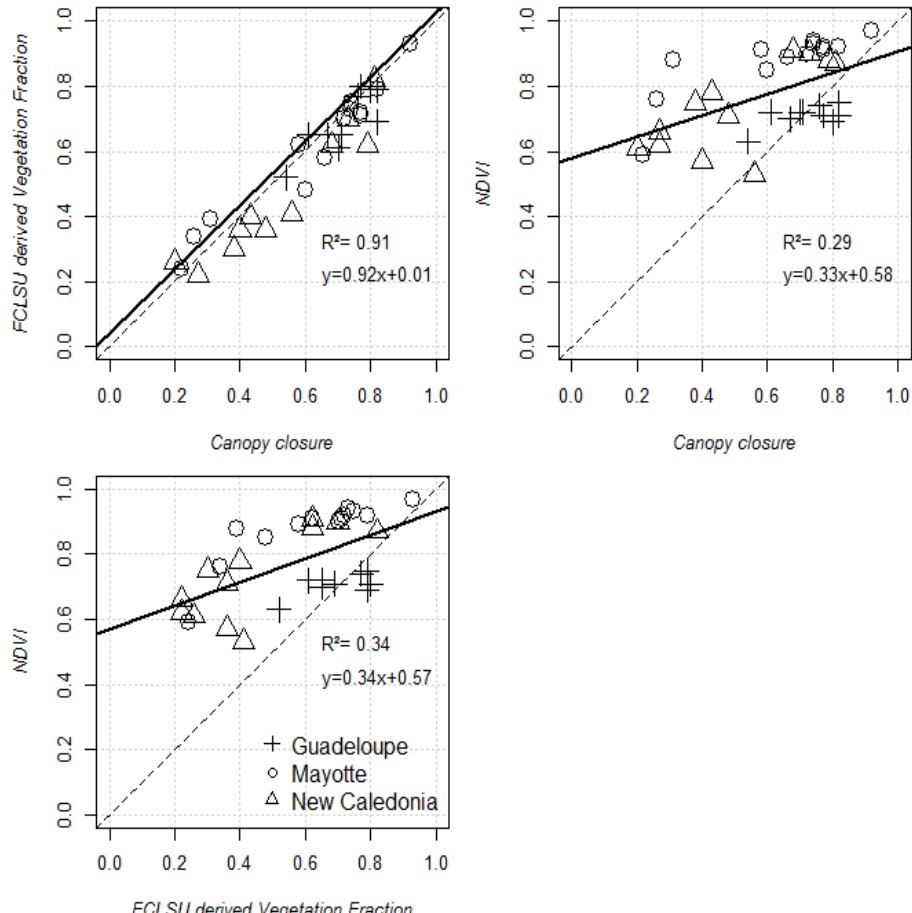

**Figure 4.** Relationships between FCLSU-derived vegetation fraction values and canopy closure estimates from hemispherical photographs (**left**) and NDVI values (**right**). The continuous line indicates the best linear model between each pair values.

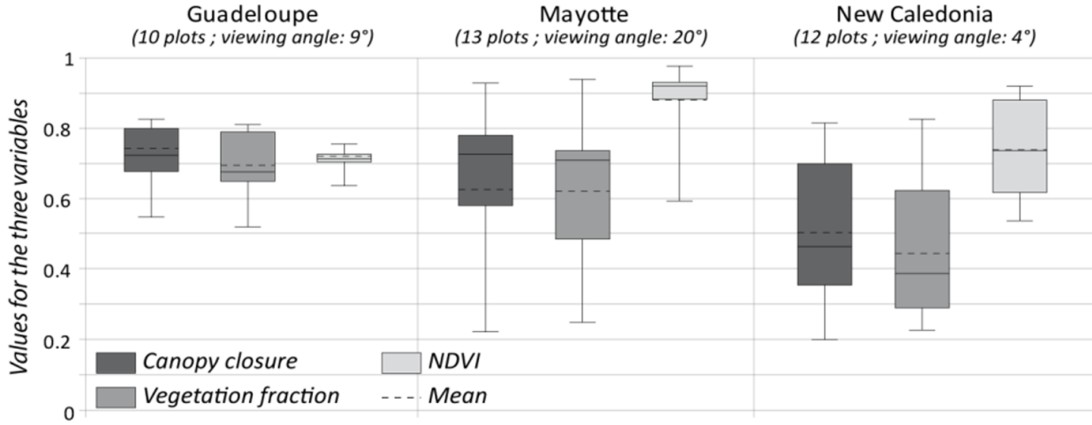

**Figure 5.** Range of variations of canopy closure, VF and NDVI values obtained from the analysis of Pleiades-1B images acquired over the regional sites. Box plots give the mean, median, quartiles, min and max values.

In Guadeloupe, the lowest variation range and the highest mean value were observed of each of the three parameters considered.

In Mayotte, the VF and canopy closure had close mean values of 0.63 and 0.62, respectively, and close variation ranges of 0.22 to 0.92 and 0.24 to 0.93, respectively. The NDVI showed a very high mean value of 0.9 with a tight box plot. The sensitivity of the FCLSU vegetation fraction estimates to canopy

closure was confirmed, as was the irrelevance of NDVI for capturing the variability of mangrove canopy structure (see the circles in Figure 4, right).

In New Caledonia, mean values reached the lowest levels for both canopy closure and vegetation fraction (0.2 and 0.22, respectively). The mean value and the height of the NDVI box plot suggest a higher sensitivity than those observed in Guadeloupe and Mayotte.

### 4.2. Potential of Fraction Classification for Mapping Mangrove Forest Structures

The contributions of the water, shadow and soil fractions were examined as a function of the vegetation fraction. Mapping fractions helps in understanding mangrove zonation (Figures 6 and 7). The classes correspond to different mangrove types interpreted from field data and are mainly spatially distributed according to a VF intensity gradient.

Mapping of pixel fractions yields vegetation classes that correspond quite closely to different mangrove structures measured during field campaigns. For the three sites, VF classification is useful to distinguish dense and canopy closed *Rhizophora* dominant vegetation from sparse and/or low *Avicennia* or *Sonneratia* open canopy vegetation. Between these two types, a gradient composed of mixed mangroves shows a decreasing VF from the landward border to the center of the mangrove areas. On the other hand, some differences can be observed between the sites: for Mayotte and Guadeloupe, classes 4, 5, and 6 have a low VF and show a high standard deviation, which is not the case in New Caledonia. This is explained by the composition of mangroves. For example, in Guadeloupe and Mayotte, polygons of classes 4, 5, and 6 include some pixels covering the crowns of tall and sparse trees and others covering the gaps between trees. In New Caledonia, classes with a low VF correspond to homogeneous low *Avicennia* shrubs with low density foliage and so the variability of the pixel values is smaller in such a structure than in other territories.

More precisely, in Guadeloupe Island the WF for *Avicennia* stands is larger than for the other classes because the trees are sparse and the soil is covered by water during the rainy season (classes 6 and 3). Mixed mangroves dominated by *Rhizophora* (MIXRHI; class 5) are located in a transition with other vegetation areas, which explains why they are not very well detected. In Mayotte, the fraction classification highlights the three types of mangrove: fringe stands composed of *Sonneratia alba*, inner mangrove stands dominated by *Rhizophora*, and landward sparse *Avicennia* and *Ceriops* stands located in the border of tannes. This classification can even distinguish the *Bruguiera* sp. stand. In New Caledonia, *Rhizophora* stands comprised both low, sparse stands and tall, closed stands. This large gradient generates confusion between all the RHIDOM classes because they are not actually so different from each other. In fact classes 1 and 2 could be merged as well as classes 3 and 4. Concerning *Avicennia*, class 5 is not so different from class 6, which explains why this class is not well detected and is confused with class 6.

For the three sites, some bias in the classification analyses resulted from a border effect explained by the size of the polygons (625 m$^2$). For fringe stands, like class 6 of *Sonneratia* in Mayotte, the encroachment of the polygons on some pixels located on the nearby open-water areas resulted in a low vegetation fraction and a high water fraction.

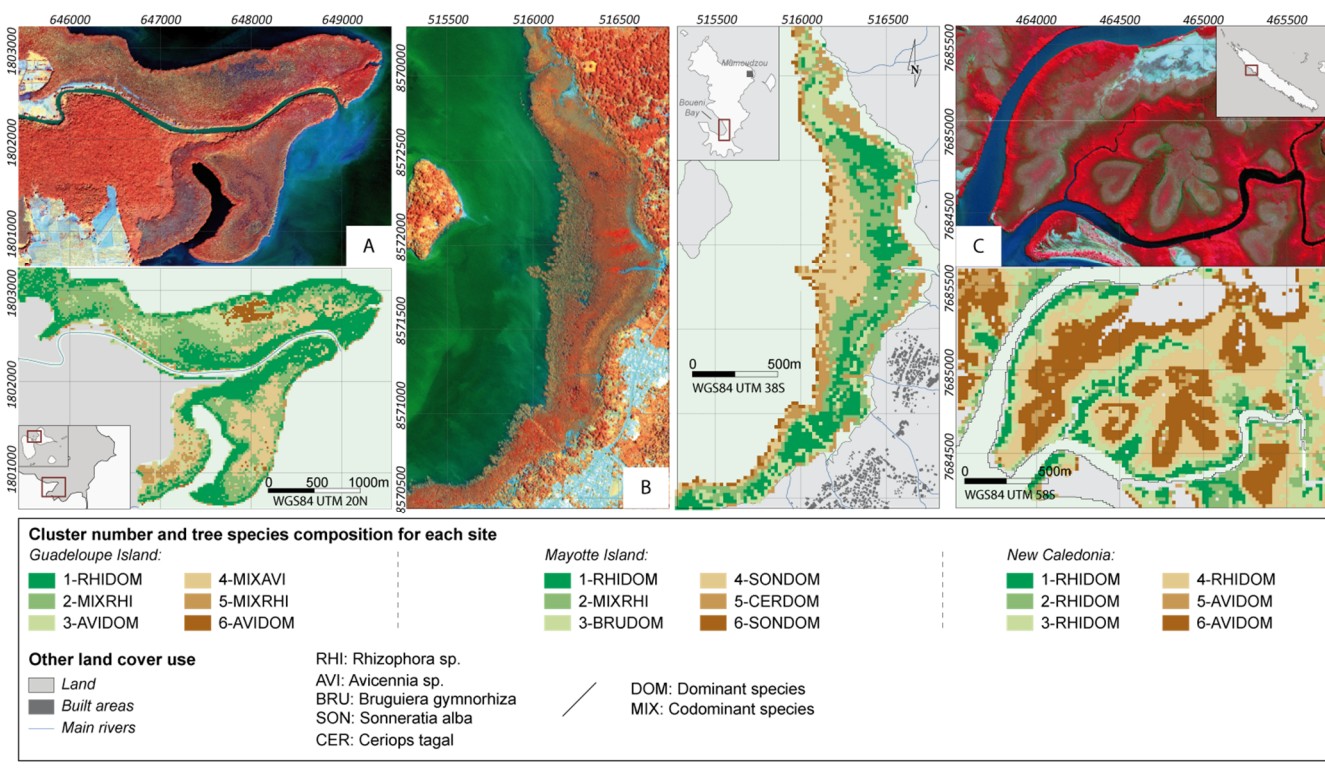

**Figure 6.** K-mean classifications (6 clusters) of mangrove vegetation structures based on fractions for the three sites.

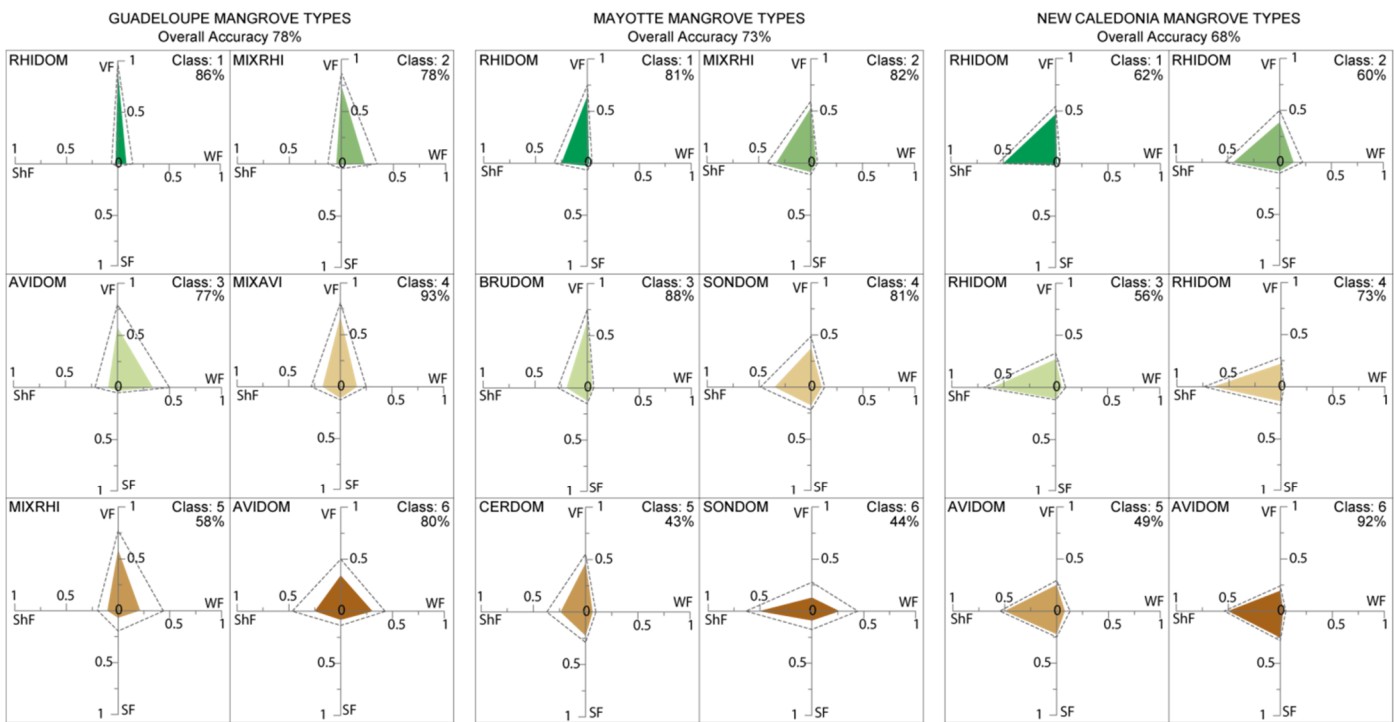

**Figure 7.** Component diagrams of the four fractions showing main mangrove vegetation structures. Coloured polygons: mean fractions for each class. Dotted line: standard deviation of the corresponding class. Percentage: user accuracy of the class.

### 4.3. Fraction versus Spectral Classification

In order to validate the fraction classification, a second K-means classification of the grid was carried out with mean spectral information. The classification results based on the mean values of spectral bands show an overall accuracy of 38.3% in Guadeloupe, 55% in Mayotte, and 63% in New Caledonia, while fraction classification shows better accuracies of 78, 73, and 68%, respectively.

However, some classes are better detected with spectral information than with fractions, such as the AVIDOM class in Guadeloupe, which reaches a user accuracy of 84% (77% with fractions), the MIXRHI class in Mayotte, which reaches 94.5% accuracy (82% with fractions), and AVIDOM in New Caledonia, which reaches 88.7% accuracy instead of 49% with fractions. This is explained by the spectral properties of mangrove species, which play a major role in discriminating mangrove types when these are composed of different species. This is not the case when the types are composed of the same species with different patterns of canopy closure. This explains why MIXAVI and AVIDOM can be clearly detected by fraction classification but not by spectral classification (because both classes are mainly composed of the same species). It also explains why the fringe stands of *Sonneratia alba* form one class with spectral classification and two classes with fraction classification. In New Caledonia, the RHIDOM mangroves are split into only three classes with spectral classification instead of four classes with fraction classification, while AVIDOM forms three classes instead of two. So, fraction classification is globally more adapted to mangrove typology based on species composition than spectral classification.

Moreover, fractions provide quantitative and standardized information that can be used to create standard typologies based only on canopy structure such as closed, medium and open mangroves stands. For example, the closed mangrove can be characterized by vegetation fractions up to 75%, no matter the location in the world or in the season (that is not the case with NDVI). This represents an advantage in using standardized typologies for national or international observatories.

## 5. Discussion and Conclusions

The reliability of FCLSU in monitoring mangrove canopy closure, and its ability to monitor canopy structure of a broad range of mangrove stands, is demonstrated by the strong correlation ($R^2 = 0.91$) found between the canopy closure estimated from field hemispherical photographs and vegetation fraction in image pixels. We also confirmed that NDVI is poorly sensitive to canopy structure in agreement with a number of previous works [59]. K-means classification of FCLSU-derived fractions also proved useful for mapping distributional patterns of mangrove zonations.

However, some limitations were encountered during this study. The exploitation of hemispherical photographs collected in the centre of each plot with lens pointing the sky works well under closed vegetation. But this HP acquisition approach is to be questioned for low or sparse vegetation. For low vegetation, HP acquisition could be done by pointing the ground combined with photograph binarisation applied to discriminate vegetation from soil. For sparse vegetation, the number of photographs must be adapted to the spatial heterogeneity.

Spectral mixture analysis of satellite images is dependent on solar illumination and viewing angles as stated in [60]. In New Caledonia, the shadow fraction is observed higher than in the other territories due probably to the combined effect of canopy gaps and low sun position (high solar zenithal angle) during the image acquisitions, this resulting in larger shadowed areas. We thus recommend prioritizing zenithal sun and viewing angles below 20° and avoiding grazing sun and hot-spot configurations as stated in [61] and illustrated through a textural analysis in [62]. Taking care of the flooding level at the image acquisition time is also crucial for evaluating the soil contribution. We also insist on the need to carry out physically-based works to interpret scattering mechanisms within forest canopies [63]. Finally, the method developed in this study is specific to mangrove forests characterized by low understory vegetation which is not the case of tall equatorial mangroves or terrestrial tropical forests for which interesting research are still to be carried out [64].

The use of the k-means classification for deriving mangrove classes based on vegetation fractions appears to be a valuable approach for analyzing and comparing vegetation structure composed of different species in different regions and distinguishing a broad range of mangrove types. Compared to a spectral-based classification approach, fraction classification shows higher overall accuracy for the three sites. This algorithm is also adapted to blind classification of mangrove types across a number of regions since the use of only six clusters showed good potential in the delineation of mangrove types, as demonstrated in Figure 6. This component diagram could then act as a learning sample in the perspective of generalizing classification over larger areas [62]. Moreover, the overall accuracy of the maps reaching 78% at best may be improved with the support of calibration or validation forest data derived from LiDAR, stereoscopic, and hyperspectral surveys.

From an ecological perspective, FCLSU advantageously proposes standardized outputs that can be compared from one site to another. For example, a clear "radial" organization consisting of *Rhizophora* stands seaward and *Avicennia* landward is observed on each of the three study sites. Very interesting is the sensitivity of our approach that varying vegetation fraction levels among and between *Sonneratia-*, *Rhizophora-*, and *Avicennia*-dominated stands suggest. This latter point may prefigure new insights in the assessment of species composition of mangrove stands from satellite data. The case study of 30–40 m high French Guiana mangroves [65] deserves our future interest for deeper understanding of FCLSU approach in a vast mangrove region where canopy height and forest structure develop from seaward to landward in a mosaic of distinct forest stands where *Avicennia* compete with *Rhizophora* species [42].

Measurement of the canopy closure using FCLSU applied to VHSR satellite images may prefigure future operational monitoring of mangroves over time and yield pivotal data for the assessment of mangrove resilience to natural or man-induced disturbances.

**Author Contributions:** Conceptualization: F.T., M.R., F.D. and C.P.; Metholodgy: F.T., M.R., F.D., C.P, F.F. and D.I.; Field data acquisition: F.T., M.R., F.F., D.I. and C.P.; Writing-review and editing: F.T., M.R. and C.P.

**Funding:** A part of this study was funded by the French Coastal Conservancy Institute. It was conducted as part of the PhD work of Florent Taureau supported by the University of Nantes.

**Acknowledgments:** The authors thank Kildine Veau and Marie Windstein for their help during the field measurements in the mangroves and Carol Robins for correcting the English of the paper.

**Conflicts of Interest:** The authors declare no conflict of interest.

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
