# Peer review of "Mapping the Mangrove Forest Canopy Using Spectral Unmixing of Very High Spatial Resolution Satellite Images"

_remotesensing, doi:10.3390/rs11030367_

Reviewer 1 Report

Dear Authors,

It was a great pleasure to read and review your paper. I can easily say that your manuscript was meticulously prepared and the findings are important. Even so, I have some questions and concerns given below;

1) Authors say that three series of digital hemispherical photographs (HP) were taken using a GoPro Hero-3. They talk about the first series obtained with a focal length of 14 mm opening at an angle of view of 149.2°. Others?

2) I have some hesitation regarding the performance of hemispherical photographs in the stands with low mean vegetation height such as < 2 m. In such stands, to measure crown closure based on hemispherical photographs may cause some uncertainties. Besides, according to my opinion, in order to determine crown closure more accurately, at least, approximately 10 photos should be taken from a plot. How did you deal with these issues?

3) I would want to see crown cover percentage instead of crown closure in the study. Crown cover percent of sample points could be easily measured from aerial photos taken from a drone. Therefore, the relationships between satellite-based variables and canopy cover percentage could be determined more reliable.

4) Did you check assumptions of linear regression analysis (e.g. normal distribution)?

5) Authors said that “A correlation analysis between the canopy closure measurement from HP and VF shows the reliability of LSU in monitoring mangrove canopy closure and its ability to monitor canopy structure on any type of mangrove”. A validation test is needed to judge this. ? A one-leave-out validation might be suitable when considering your data set.

6) I could not understand the value of Figure 7.  In my opinion, it can be deleted.

7) I found some results too detailed. If possible, lines between 298 and 333 can be summarized.

8) There is little discussion in the paper. The results should be discussed with relevant literature. For example, the paper entitled “Linear transformation to minimize the effects of variability in understory to estimate percent tree canopy cover using RapidEye data” used similar approaches to predict canopy cover percentage in a conifer forest.

Author Response

1)      Authors say that three series of digital hemispherical photographs (HP) were taken using a GoPro Hero-3. They talk about the first series obtained with a focal length of 14 mm opening at an angle of view of 149.2°. Others?

Our answer: The same focal length was used for all photographs. It was an error. The original text has been modified as follows:

L162 – L 163: Hemispherical photographs were obtained with a focal length of 14 mm opening at an angle of view of 149.2°.

2)      I have some hesitation regarding the performance of hemispherical photographs in the stands with low mean vegetation height such as < 2 m. In such stands, to measure crown closure based on hemispherical photographs may cause some uncertainties. Besides, according to my opinion, in order to determine crown closure more accurately, at least, approximately 10 photos should be taken from a plot. How did you deal with these issues?

Our answer: Approximatively 15 to 20 photographs were taken for each plot with a one second delay between each photo in order to have same exposition condition for each photo. All series for each plot were taken at the same emplacement: in the center of the plot. We agreed with the reviewer to say that different series of photographs taken at different location in the plot (for example one in the center, and four at each corner) could improve performance of canopy closure estimations especially in low or sparse vegetation condition. Please have a look to the ‘data and method’ and ‘discussion’ sections where we tried to improve both methods and discussion sections as follows:

L164 – L169: All photographs were taken from the forest plot center. Approximatively 15 to 20 photographs were taken for each plot with a one second delay between each photo in order to have same light exposition condition for each photo. Then the best photograph i.e. the photograph which is not affected by sunlight saturation or branches movement caused by wind has been kept and used for analysis.

L397 – L402: The exploitation of hemispherical photographs collected in the centre of each plot with lens pointing the sky works well under closed vegetation. But this HP acquisition approach is to be questioned for low or sparse vegetation. For low vegetation, HP acquisition could be done by pointing the ground combined with photograph binarisation applied to discriminate vegetation from soil. For sparse vegetation, the number of photographs must be adapted to the spatial heterogeneity.

3)      I would want to see crown cover percentage instead of crown closure in the study. Crown cover percent of sample points could be easily measured from aerial photos taken from a drone. Therefore, the relationships between satellite-based variables and canopy cover percentage could be determined more reliable.

Our answer: The two approaches are very different and this difference is caused by the level of analysis we choose to work with, and which is related to a pixel-to-pixel analysis or a sub-pixel analysis. In the case of using crown cover we chose a binary approach, and for each control point interpreted on the aerial photograph is assigned a binary attribute (e.g. vegetated/not-vegetated). The main limitation of this approach is encountered in very dense vegetation. How the points will be interpreted with for example a 75% canopy closure? The interpreter will probably classified all the point as vegetated because it is not possible to see the space between branches, crowns, shadows etc. But in reality, those spaces and shadows will affect the spectrum values of pixels composing satellite images which can be underline with spectral mixture analysis. That’s mean that a point occurring in the middle of a crown is not necessary 100% vegetated.

On the other hand, canopy closure approach seems to be limited with sparse or low vegetation. So, in my opinion, the adapted method should be used according the structural properties of the vegetation studied. For example, in a savanna vegetation where we can found dense isolated trees separated with bare ground, the canopy cover can be used, but in order to analyze a variable vegetation continuum it is better to use canopy closure.

4)      Did you check assumptions of linear regression analysis (e.g. normal distribution)?

Our answer: The distribution of canopy closure estimated from hemispherical photographs (explained variable) is not normal. The Shapiro-Wilk test give W=0.89 with a significant p-value at 0.0031 with a given error at 5%. The Q-Q plot also shows that the distribution is not normal.

But more important than the normal distribution of the explained variable is the normality of the residues between the explained variable and modelled values. This latter has been positively checked with a Shapiro-Wilk test (n<50) and proves that our model is reproducible and predictive (please see Results section). We then assumed that parametric tests usable for the study.

5)      Authors said that “A correlation analysis between the canopy closure measurement from HP and VF shows the reliability of LSU in monitoring mangrove canopy closure and its ability to monitor canopy structure on any type of mangrove”. A validation test is needed to judge this. ? A one-leave-out validation might be suitable when considering your data set.

Our answer: Yes. We adjusted our text in agreement with the reviewer's comment. The text is now

L360 – L363: The reliability of FCLSU in monitoring mangrove canopy closure, and its ability to monitor canopy structure of a broad range of mangrove stands, is demonstrated by the strong correlation (R² = 0.91) found between the canopy closure estimated from field hemispherical photographs and vegetation fraction in image pixels.

6)      I could not understand the value of Figure 7.  In my opinion, it can be deleted.

Our answer: We judge the figure useful for highlighting the main contributors of spectral signatures from a broad range of forest stands (see our answer to previous comment). But you are right the figure deserve to be more referred and more precision have been added in the text. Please, see particularly our improvements:

L310 – L318: Between these two types, a gradient composed of mixed mangroves shows a decreasing VF from the landward border to the centre of the mangrove areas. On the other hand, some differences can be observed between the sites: for Mayotte and Guadeloupe, classes 4, 5 and 6 have a low VF and show a high standard deviation, which is not the case in New Caledonia. This is explained by the composition of mangroves. For example, in Guadeloupe and Mayotte, polygons of classes 4, 5 and 6 include some pixels covering the crowns of tall and sparse trees and others covering the gaps between trees. In New Caledonia, classes with a low VF correspond to homogeneous low Avicennia shrubs with low density foliage and so the variability of the pixel values is smaller in such a structure than in other territories.

7)      I found some results too detailed. If possible, lines between 298 and 333 can be summarized.

Our answer: Done. Please refer to the new section 4.2.

8)      There is little discussion in the paper. The results should be discussed with relevant literature. For example, the paper entitled “Linear transformation to minimize the effects of variability in understory to estimate percent tree canopy cover using RapidEye data” used similar approaches to predict canopy cover percentage in a conifer forest.

Our answer: Please refer to the new conclusion and discussion section (L367 – L 383) where we made some modifications accordingly to your suggestion.

Reviewer 2 Report

The manuscript is well written and objectives were well supported by data and analysis. Few points may be added to improve the manuscript.   

1.      In Fig 1 caption, authors should write what satellite images were used.

2.      The spatial and spectral resolutions of Pleiades-1B should be added in method section. 

3.      In Fig.6, Clusters 1 to 6 may be expanded in the caption itself to know the dominant sps so that reader may not require to check with Table 1.

4.      The corresponding areas of dominant sps in three sites may be given in Tabular format.

5.      The limitations of the methodology may be written if any under discussion.

Author Response

Responses to Reviewer 2's comment:

1)      In Fig 1 caption, authors should write what satellite images were used.

Our answer: Done. The new figure 1 caption is: ‘Study sites and location of plots in the three islands (French Overseas Territories) based on Pleiades 1B images’.

2)      The spatial and spectral resolutions of Pleiades-1B should be added in method section.

Our answer: Done. Please refer to table 2.

3)      In Fig.6, Clusters 1 to 6 may be expanded in the caption itself to know the dominant sps so that reader may not require to check with Table 1.

Our answer: Yes. We enlarged the figure caption to integrate the dominant species for the 6 mangrove classes.

4)      The corresponding areas of dominant sps in three sites may be given in Tabular format.

Our answer: Yes, the species distribution deserves to be clarified. We choose to indicate for the three sites the dominant species and the colors to which they stand for in the caption. Please refer to the figure 6 and the new caption as previously mentioned in our answer to comment 3.

5)      The limitations of the methodology may be written if any under discussion.

Our answer: Please refer to the new last section.

L367-L383: However, some limitations were encountered during this study. The exploitation of hemispherical photographs collected in the centre of each plot with lens pointing the sky works well under closed vegetation. But this HP acquisition approach is to be questioned for low or sparse vegetation. For low vegetation, HP acquisition could be done by pointing the ground combined with photograph binarisation applied to discriminate vegetation from soil. For sparse vegetation, the number of photographs must be adapted to the spatial heterogeneity.

Spectral mixture analysis of satellite images is dependent on solar illumination and viewing angles as stated in [60]. In New Caledonia, the shadow fraction is observed higher than in the other territories due probably to the combined effect of canopy gaps and low sun position (high solar zenithal angle) during the image acquisitions, this resulting in larger shadowed areas. We thus recommend prioritizing zenithal sun and viewing angles below 20° and avoiding grazing sun and hot-spot configurations as stated in [61] and illustrated through a textural analysis in [62]. Taking care of the flooding level at the image acquisition time is also crucial for evaluating the soil contribution. We also insist on the need to carry out physically-based works to interpret scattering mechanisms within forest canopies [63]. Finally, the method developed in this study is specific to mangrove forests characterized by low understory vegetation which is not the case of tall equatorial mangroves or terrestrial tropical forests for which interesting research are still to be carried out [64].